# Replication of Human Sapovirus in Human-Induced Pluripotent Stem Cell-Derived Intestinal Epithelial Cells

**DOI:** 10.3390/v15091929

**Published:** 2023-09-15

**Authors:** Naomi Matsumoto, Shiho Kurokawa, Shigeyuki Tamiya, Yutaka Nakamura, Naomi Sakon, Shoko Okitsu, Hiroshi Ushijima, Yoshikazu Yuki, Hiroshi Kiyono, Shintaro Sato

**Affiliations:** 1Department of Virology, Research Institute for Microbial Diseases, Osaka University, Osaka 565-0871, Japan; 2Department of Human Mucosal Vaccinology, Chiba University Hospital, Chiba 260-8670, Japan; 3Department of Microbiology and Immunology, School of Pharmaceutical Sciences, Wakayama Medical University, Wakayama 640-8156, Japan; 4Department of Microbiology, Osaka Institute of Public Health, Osaka 537-0025, Japan; 5Division of Microbiology, Department of Pathology and Microbiology, Nihon University School of Medicine, Tokyo 173-8610, Japan; 6Future Medicine Education and Research Organization, Chiba University, Chiba 263-8522, Japan; 7CU-UCSD Center for Mucosal Immunology, Allergy, and Vaccines (cMAV), Departments of Medicine and Pathology, University of California, San Diego, CA 92093-0956, USA

**Keywords:** human sapovirus, intestinal epithelial cells, in vitro replication, histo-blood group antigens, induced pluripotent stem cell

## Abstract

Sapoviruses, like noroviruses, are single-stranded positive-sense RNA viruses classified in the family *Caliciviridae* and are recognized as a causative pathogen of diarrhea in infants and the elderly. Like human norovirus, human sapovirus (HuSaV) has long been difficult to replicate in vitro. Recently, it has been reported that HuSaV can be replicated in vitro by using intestinal epithelial cells (IECs) derived from human tissues and cell lines derived from testicular and duodenal cancers. In this study, we report that multiple genotypes of HuSaV can sufficiently infect and replicate in human-induced pluripotent stem cell-derived IECs. We also show that this HuSaV replication system can be used to investigate the conditions for inactivation of HuSaV by heat and alcohol, and the effects of virus neutralization of antisera obtained by immunization with vaccine antigens, under conditions closer to the living environment. The results of this study confirm that HuSaV can also infect and replicate in human normal IECs regardless of their origin and are expected to contribute to future virological studies.

## 1. Introduction

Sapoviruses are single-stranded positive-sense RNA viruses that belong to the family *Caliciviridae* and, like noroviruses, which also belong to the same family, are known as diarrheal viruses. Sapovirus infections are usually sporadic and are mild or asymptomatic compared to norovirus infections, but outbreaks occur in infant and elderly care facilities and occasionally become severe and even fatal [1,2]. It has also been reported that the percentage of HuSaV-infected patients has increased since the lifting of behavioral restrictions due to the coronavirus disease 2019 pandemic [3,4,5,6]. Sapoviruses are currently classified into 19 genogroups based on the sequence of the structural protein VP1 [7], of which GI, GII, GIV, and GV are known as human sapoviruses (HuSaVs) that infect humans [8]. HuSaV has been further subdivided into 18 genotypes. Recently, Oka et al. reported a PCR assay system capable of detecting all genotypes of HuSaV [9].

In vitro replication and propagation of human norovirus, which is morphologically and genomically closely related to HuSaV, has recently become possible using intestinal epithelial cells (IECs) derived from human tissues or human induced pluripotent stem cells (iPSCs) [10,11]. Although HuSaV has been shown to be unable to replicate in IECs derived from human tissues [12], virus replication was shown to increase up to ~6 log_10_-fold in vitro using an IEC line (HuTu-80) derived from human duodenal carcinoma and bile acid, allowing measurement of HuSaV infectivity [13]. The use of this culture system is expected to facilitate further analysis of the virological properties of HuSaVs; however, for further advancing our knowledge of the virus using the culture system resembling human intestine, it is necessary to construct additional a HuSaV growth system ideally using normal cells rather than cancer-derived cells.

Accordingly, in this study, we investigated whether IECs derived from human iPSCs could be used to replicate HuSaV in vitro and thus provide additional culture system for our continuous molecular and cellular understanding of HuSaV infection.

## 2. Materials and Methods

### 2.1. Cells

The human iPSC-lines TkDN4-M and TkD2 were established using the same protocol as previously reported [14] and then provided by The Institute of Medical Science, The University of Tokyo. Culture and passage of IECs differentiated from human iPSC lines were performed as described previously [11,15,16]. Conditioned medium containing mouse Wnt3a, human R-spondin1, and human Noggin or R-spondin1 and Noggin was prepared as described previously [16]. For HuSaV infection, IECs were dissociated and seeded on 2.5% Matrigel-coated 96-well plates at 2 × 10^4^ cells/well in 100 μL organoid culture medium (Advanced Dulbecco’s modified Eagle medium/F12 [Thermo Fisher Scientific, Waltham, MA, USA] supplemented with 10 mM HEPES [pH 7.3; Thermo Fisher Scientific]; 2 mM Glutamax [Thermo Fisher Scientific]; 100 U/mL penicillin; 100 μg/mL streptomycin; 25% conditioned medium containing mouse Wnt3a, human R-spondin1, and human Noggin; 1× B-27 [Thermo Fisher Scientific]; 50 ng/mL mouse epidermal growth factor [Peprotech, Cranbury, NJ, USA]; 50 ng/mL human hepatocyte growth factor [R&D Systems, Minneapolis, MN, USA]; 10 μM SB202190 [Sigma-Aldrich, St. Louis, MO, USA]; and 500 nM A83-01 [Tocris, Bristol, UK]) plus 10 μM Y-27632 [Fujifilm Wako, Osaka, Japan]. After 2 days of culture in a 5% CO_2_ incubator at 37 °C, the medium was changed to differentiation medium (Advanced Dulbecco’s modified Eagle medium/F12 supplemented with 10-mM HEPES; 2 mM Glutamax; 100 U/mL penicillin; 100 μg/mL streptomycin; 1× B-27; 12.5% conditioned medium containing human R-spondin1, and human Noggin; 50 ng/mL mouse epidermal growth factor; and 500 nM A83-01). After another 2 days, the medium was changed to differentiation medium with or without 0.03% porcine bile (Sigma-Aldrich). The cells were incubated for another 2 days and then used for subsequent experiments.

### 2.2. Screening and Genotyping of HuSaV

Stool samples were collected from infants and children with acute gastroenteritis who visited three pediatric outpatient clinics in Gunma, Kyoto, and Shizuoka prefecture, Japan. All samples were screened for enteric viruses including HuSaV as described previously [17]. To determine HuSaV genotype, we amplified VP1 cording region by reverse transcription-polymerase chain reaction by using primers SLV5317 and SLV5749 [18,19]. Nucleotide sequence of the PCR products was determined and then compared with the available data in the GenBank database using the Basic Local Alignment Search Tool (https://blast.ncbi.nlm.nih.gov/Blast.cgi (accessed on June 2016–September 2018 or November 2019–October 2020)). Some genotypes of HuSaV used in this study have already been reported [17], and their genomic information has been registered and assigned accession numbers (see Table 1).

### 2.3. Preparation of Plasmid as Standard for HuSaV Genome Copies

To prepare plasmid as standard for quantification of HuSaV copy numbers, the viral genome was extracted from GI.1 HuSaV (sample ID 18002). A region of approximately 2.4 kbp containing the junction of ORF1 and ORF2 was amplified by RT-PCR and subsequent PCR using forward primer (5′-TWTGAYYWGGCYCTCGCCACCTACRA-3′) and Rv primer (5′-CCACACGCGTTCGGGTGGTTAAATGTG-3′). The PCR product was cloned into the pCR4 vector (Invitrogen). The concentration of the purified plasmid was determined by measuring the optical density at 260 nm. A total of 5.39 ng of this plasmid was used as 10^9^ copy equivalents of the HuSaV genome.

### 2.4. HuSaV Preparation and Infection

HuSaV-positive stool samples were suspended in phosphate-buffered saline at 10% (*w*/*v*) by vigorous vortexing. The suspensions were centrifuged at 12,000× *g* for 30 min, and the supernatants were serially filtered with 0.45- and 0.22-μm filters. The filtered samples were aliquoted and stored at −80 °C as undiluted virus solutions (Table 1). Immediately before use, each virus solution was diluted with the base medium (Advanced Dulbecco’s modified Eagle medium/F12 supplemented with 10-mM HEPES; 2 mM Glutamax; 100 U/mL penicillin; and 100 μg/mL streptomycin). The prepared IECs (3–6 wells/sample) were inoculated with 100 μL (2 × 10^4^–2 × 10^8^ genome equivalents; 6.25 × 10^4^–6.25 × 10^8^ genome equivalents/cm^2^) diluted virus solutions and then left for 1 h in a 5% CO_2_ incubator at 37 °C. The inoculum was then removed, and the cells were washed twice with 150 μL base medium. One hundred microliters of differentiation medium with or without 0.03% bile was added to the cells, which were then pipetted lightly twice and collected. This step was performed again, and the samples were collected as 1 h post-infection (hpi) reference samples (total 200 μL). Another 100 μL differentiation medium with or without 0.03% bile was added to each well, and the mixtures were then cultured for 24–96 h in a 5% CO_2_ incubator at 37 °C. The supernatants were then collected with one wash, in the same way as 1 hpi reference samples (total 200 μL).

### 2.5. Inactivation of HuSaVs

#### 2.5.1. Heat Inactivation of HuSaVs

GI.2 and GII.1 HuSaVs (2 × 10^6^ genome equivalents) were incubated at 85 °C for 0, 2, or 5 min. To stop the heat treatment, the viruses were placed on ice for 5 min. Each virus was inoculated onto prepared IECs with 0.03% bile as described above.

#### 2.5.2. Alcohol Inactivation of HuSaVs

One volume of GI.2 and GII.1 HuSaVs (2.1 × 10^7^ and 1.0 × 10^8^ genome equivalents, respectively) was mixed with three volumes of 70% ethanol-0.05% magnesium sulfate with or without 1% citric acid and allowed to sit at room temperature for 5 min. Then, each suspension was subjected to a 100-fold dilution with base medium (Advanced Dulbecco’s modified Eagle medium/F12 supplemented with 10 mM HEPES [pH 7.3], 2 mM Glutamax, and 100 units/mL penicillin plus 100 μg/mL streptomycin). Each treated virus (5.3 × 10^6^ [GI.2] or 2.5 × 10^7^ [GII.1] genome equivalents) was inoculated onto prepared IECs with 0.03% bile as described above. It has been reported previously that the disinfectants used in this study are not cytotoxic to IEC#17 [15].

#### 2.5.3. Neutralization Assay of HuSaVs

GII.1 and GII.3 HuSaVs (2 × 10^6^ genome equivalents) were pretreated with 0.2 μL antiserum prepared from GII.1 HuSaV VLP-immunized or non-immunized mice at 37 °C for 90 min. Each virus was inoculated onto prepared IECs with 0.03% bile as described above.

### 2.6. Quantification of Virus Genome Equivalents

A PureLink Viral RNA/DNA Mini Kit (Invitrogen) was used to prepare RNA from diluted virus solutions and collected samples. Complementary DNA of the HuSaV genomic RNA was generated by using a PrimeScript™ RT reagent Kit (TaKaRa, Shiga, Japan) and then was quantified by real-time PCR with Premix Ex Taq™ (Probe qPCR) (TaKaRa), primers (SaV124F, 1F, and 1245R), FAM-labeled MGB probe (SaV124TP) [20], and LightCycler 480 System (Roche, Basel, Switzerland) or QuantStudio 5 Real-Time PCR System (Thermo Fisher) following the manufacturer’s protocols. The PCR reaction was performed at 95 °C for 15 s followed by 45 cycles of 94 °C for 15 sec and 62 °C for 60 s.

### 2.7. Lentiviral Infection

Lentiviral expression plasmid for the *FUT2* gene was constructed by inserting full-length PCR fragments into CSII-EF-MCS-IRES2-Venus using Lipofectamine 2000 transfection reagent (Thermo Fisher Scientific). *FUT2* insertion plasmid or empty plasmid was transfected into HEK293T cells according to the supplier’s protocol, together with packaging plasmids (pCAG-HIVgp) and VSV-G- and Rev expression plasmids (pCMV-VSV-G-RSV-Rev). After 6 h, the medium was replaced with fresh medium containing 10 µM forskolin, and further 48 h later, the supernatant containing lentivirus was collected and filtered. The plasmids for lentivirus production were kindly provided by Dr. Hiroyuki Miyoshi through the RIKEN BioResource Center, Japan. IEC#34 organoids were dissociated and seeded on 2.5% Matrigel-coated 6-well plates at 2 × 10^5^ cells/well in 2 mL organoid culture medium. After 24 h, cells were infected with the lentiviral solution containing 10 μg/mL polybrene using centrifugation (800× *g* for 90 min). After washing with PBS three times, cells were harvested with TrypLE Express and embedded in Matrigel to regenerate organoids.

### 2.8. Production of SaV VLP

The nucleotide sequence encoding VP1 of HuSaV GII.1 (sample ID 18003) was sequenced and cloned into the pFastBac1 vector (Invitrogen, Waltham, MA, USA). Each construct transformed DH10Bac competent cells (Invitrogen) and recombinant bacmid was purified. Recombinant baculovirus production was obtained by transfection of recombinant bacmid into ExpiSf9 cells using the ExpiSf Expression system (Thermo Fisher Scientific) and collecting the supernatant on day 5–6 of infection. Inoculating ExpiSf9 cells with this viral stock, the culture supernatant was harvested to confirm expression. The supernatant was centrifuged at 12,000× *g* for 10 min, and after repeated centrifugation, the supernatant was passed through a 0.45 μm filter (Merk Millipore, Burlington, MA, USA). After centrifugation at 113,000× *g* for 4 h, the precipitate was suspended in TNE buffer (10 mM Tris, 100 mM NaCl, 1 mM EDTA [pH 7.4]) and left at 4 °C overnight. The samples were then suspended in 42% CsCl-TNE buffer and subjected to cesium chloride equilibrium density centrifugation at 139,000× *g* for 20 h, after which each fraction was collected and dialyzed against TNE buffer. The final sample was negatively stained with 1% uranyl acetate solution, and the VLP structure was confirmed by transmission electron microscopy (100 kV, JEM-1400, JEOL, Tokyo, Japan).

### 2.9. Generation of HuSaV GII.1 Antiserum

Purified HuSaV VLP (10 μg) and Imject Alum (Thermo Scientific) were intramuscularly injected into five BALB/cAJc1 (CLEA Japan, Shizuoka, Japan) mice, and 2 weeks later, 10 μg of VLP was again intramuscularly injected for immunization. Blood samples collected at the third week of immunization were subjected to ELISA as described below and determine antibody titer. ELISA has been done as follows; 96 well plates coated with 100 ng/well of SaV VLP the day before were washed 5 times with PBST (0.05% Tween20-PBS). Blocking was performed with 1% BSA-PBST at room temperature for 1 h. After 3 washes, 100 μL/well of serum, 2-fold step-diluted in 1% BSA-PBST, was incubated for 1 h at room temperature. Following 5 washes, Goat anti-Mouse IgG-HRP (SouthernBiotech, Birmingham, AL, USA) diluted 4000-fold in PBST was incubated for 1 h at room temperature. The plate was washed 5 times, and 100 μL/well of KPL TMB Peroxidase Substrate (LGC seracare, Milford, MA, USA) was added, followed by the addition of 2 N sulfuric acid to stop the reaction. The 450 nm absorbance was measured with an iMark microplate reader (BIO-RAD, Hercules, CA, USA) and analyzed using Microplate Manager 6.2. Antisera from five mice were mixed in equal amounts and used in the experiment.

### 2.10. Statistical Analysis

Results were compared using unpaired two-tailed Student’s *t*-tests. Differences were considered significant if the *p* value was less than 0.05. All statistical analyses were conducted using GraphPad Prism 8 and 9.

### 2.11. Ethics Statement

After obtaining informed consent from participants, all stool samples were collected and provided by the Nihon University School of Medicine. The study was approved by the Human Ethical Committees of Osaka University (approval no. 28-3), Wakayama Medical University (approval no. T3566 and T3716), and Nihon University (approval no. 29-9 and 2022-05).

## 3. Results

Because the addition of bile (or bile acids) has been reported to be necessary for the in vitro replication of HuSaVs using the HuTu-80 cell line [13], we first attempted to infect human iPSC-derived IECs cultured in the presence of bile with 19 HuSaVs. As a result, we were able to confirm the replication of virus genotypes GI.1, GI.2, GII.1, GII.3, and GIV.1 in IECs previously established from the TkDN4-M human iPSC-line, IEC#17 (Table 1). We then evaluated whether bile was required for replication of the virus samples, and our results showed that the addition of bile was not always necessary (Figure 1). However, like human norovirus replication in our system [11], bile supplementation appeared to enhance the replication of HuSaV. Therefore, all subsequent experiments were performed using cells cultured in the presence of bile. Of note, GI.2 (sample ID 18429) and GII.1 (sample ID 18003) showed a more than 100-fold increase in replication in our experimental system.

To determine the viral copy number limits for the in vitro replication of these two virus samples of GI.2 and GII.1, the fecal suspensions were serially diluted 10-fold before use in the experiment. Notably, both viruses required more than 2 × 10^6^ copies/100 µL for replication (Figure 2A). Next, we examined the viral replication kinetics of the two virus samples and found that they seemed to reach a plateau at 96 hpi when IECs were infected with 2 × 10^7^ copies/100 µL (Figure 2B). This kinetics was quite similar to the in vitro replication of human norovirus, as described previously [10,11]. Accordingly, these findings suggested that HuSaV could infect and replicate in TkDN4-M-derived IECs (IEC#17) in the same manner as the human norovirus.

The GII.4 genotype of human norovirus is currently the most prevalent genotype worldwide [21]. Glycosylation of α(1,2)-type fucose is required for binding of GII.4 norovirus into host cells [22] and subsequent intracellular replication [10]. Fucosylation of IECs is carried out by fucosyltransferase 2 (FUT2), and when this enzyme is deleted or inactivated, infection and subsequent replication of GII.4 norovirus is not observed [10]. Therefore, we next examined whether HuSaV replication in our in vitro growth system required glycosylation of α(1,2)-type fucose on IECs. To this end, we prepared a new IEC line (IEC#34) from TkD2 cells, a human iPSC line harboring an inactive mutation in the *FUT2* gene. *Ulex europaeus* agglutinin-1 (UEA-1), which binds specifically to α(1,2)-type fucose, reacted well with IEC#17; however, no UEA1-positive cells were found in IEC#34 (Appendix A). When the human *FUT2* gene was introduced into IEC#34 using a lentiviral vector, UEA1-positive cells were observed (Appendix A). Consistent with this, replication of GII.4 human norovirus was observed in IEC#17 [11,15] and *FUT2*-introduced IEC#34 but not in mock introduced IEC#34 (Appendix A). From these results, we concluded that IEC#34 was an IEC line harboring a non-secretory-type mutation, and its phenotype could be rescued by expression of an exogenous *FUT2* gene. When we confirmed the multiplication of HuSaV in these cells, we found that viruses of all examined genotypes were able to grow in nonsecretory-type IEC#34 (Figure 2C). The growth of these viruses was not affected by *FUT2* gene expression. These results indicate that the GI.2, GII.1, and GII.3 genotypes of HuSaV do not require α(1,2)-type fucose for host entry and replication, unlike the GII.4 genotype of human norovirus.

Next, we examined the conditions for HuSaV inactivation using the established in vitro HuSaV replication system. To investigate inactivation by heating, we incubated the viral solution at 85 °C for 2 or 5 min before infecting the cells. Our results showed that HuSaVs with the GI.2 or GII.1 genotype showed loss of replication ability after 5 min of heating (Figure 3A). We have previously reported that alcohol solutions with pH values outside the neutral range, particularly acidic alcohols, can sufficiently inactivate the replication ability of human norovirus [15]. A similar test was conducted for HuSaV; three volumes of 70% ethanol with or without 1% citric acid were mixed with 1 volume of virus suspension and allowed to sit at room temperature for 5 min. The experiment was conducted in the presence of 0.05% magnesium sulfate to reduce the effects of organic matter from the fecal suspension. As in the case of human norovirus, 70% ethanol had no effect on the replication capacity of HuSaV (Figure 3B). By contrast, acidic ethanol substantially inactivated both genotypes of HuSaV. These results suggest that the susceptibility of HuSaV to heat-induced inactivation and alcohol treatment is similar to that of human noroviruses.

Vaccine development against human norovirus using virus-like particles (VLPs) as antigens is in progress, and it has been reported that vaccination with VLPs produces antibodies that can neutralize norovirus and thus prevent infection [11,23,24]. Using our newly developed HuSaV in vitro replication system, we investigated whether HuSaV VLPs could serve as vaccine antigens that could induce neutralizing antibodies. We generated a bacmid incorporating a cDNA encoding VP1 of HuSaV GII.1 (sample ID 18003) and used an insect cell expression system to generate HuSaV GII.1 VLPs (Appendix A). The resulting VLPs were immunized in mice to obtain anti-GII.1 VLP antiserum (Appendix A).

When GII.1 HuSaVs were pre-incubated with 0.2 μL of anti-HuSaV GII.1 antiserum, replication was substantially inhibited (Figure 4). However, the antiserum failed to inhibit the replication of the other genotype of GII HuSaV, GII.3, in our in vitro replication system. These results suggest that VLPs are suitable as vaccine antigens against HuSaV infection, although cross-reactivity between different genotypes is not expected.

## 4. Discussion

Like norovirus, HuSaV is a diarrheal virus that infects a wide range of people from infants to adults. Although HuSaV infection is relatively milder than norovirus infection and often resolves without the need of clinical care [1], HuSaV infection can cause severe illness and even death in the infants and elderly with a weakened immune system [25]. Therefore, vaccines and treatments against HuSaV are needed, especially immunocompromised populations. However, the lack of a reliable in vitro culture system resembling the human intestinal epithelium for HuSaV has been one of major obstacles for the development of vaccines and therapeutic drugs.

In vitro cultivation of human noroviruses is currently established by using IECs prepared from human tissues or human iPSCs [10,11]; however, in vitro growth of the HuSaV was not observed in IECs derived from human tissues [12]. By contrast, a recent report showed that HuSaV could replicate in vitro in HuTu-80 cells derived from human duodenal carcinoma [13]. In the current study, we demonstrated that HuSaV could replicate in IECs derived from human iPSCs (Table 1 and Figure 2A). Because these cells are normal, noncancerous IECs, we believe that HuSaV characterization and inactivation assays can be performed in a more physiological environment than that of the tumor-derived IECs with the HuTu-80 cell line. In fact, the replication kinetics of HuSaV appeared to plateau at 96 hpi (Figure 2B), similar to that observed for human norovirus [10,11]. Furthermore, the addition of bile increased the replication efficiency of HuSaV in our in vitro replication system, but it was not a requirement (Figure 1).

Our in vitro replication system can also be applied to viral inactivation studies. Both GI.2 and GII.1 HuSaVs were found to be inactivated by heating at 85 °C for 5 min. Interestingly, however, GI.2 HuSaV was substantially inactivated by heating at 85 °C for 2 min, while only insufficient inactivation was observed for GII.1 HuSaV (Figure 3A). Takagi et al. reported that GII.3 HuSaV is more resistant than GI.1 HuSaV to 60 °C [13]. Taken together, it might be possible that GII genogroup HuSaVs are more resistant to heating than GI genogroup HuSaVs. In addition, we showed for the first time that HuSaV, similar to human norovirus, was resistant to alcohol treatment but could be substantially inactivated by exposure to acidic alcohols (Figure 3B). Furthermore, it was indicated that immunization of mice with HuSaV VLPs together with adjuvants induced neutralizing Abs (Figure 4), suggesting that HuSaV VLPs are good candidates for vaccine antigens like human norovirus VLPs [11]. We have previously reported that polyclonal Abs obtained by immunizing rabbits with VLPs of the GII.17 genotype norovirus have cross-reactivity to the GII.4 genotype norovirus, but not vice versa [11]. In addition, the anti-GII.17 norovirus Abs did not show cross-reactivity against GII.3 and GII.6 noroviruses. In the current study, we were able to confirm that the antiserum obtained by immunization with VLPs of GII.1 HuSaV had neutralizing activity but showed no cross-reactivity against GII.3 HuSaV belonging to the same genogroup. Taken together, these findings suggest that it may be difficult to develop a vaccine that can induce neutralizing Abs with cross-reactivity against different genotype viruses, both noroviruses and HuSaVs.

The severity of diarrheal symptoms induced by HuSaV infection is generally thought to be milder than that of norovirus infection [1,4]. Therefore, further studies are necessary to clarify the differences in biology and replication mechanisms between HuSaVs and human noroviruses using in vitro growth systems with the same cells. To this end, our human iPSC-derived IECs provide an opportunity to perform head-to-head comparisons between the two virus strains in the family *Caliciviridae*.

α(1,2) fucose, which is added to the apical surface of IECs by FUT2, is essential for infection by the GII.4 genotype of norovirus [10]. Indeed, GII.4 norovirus was unable to replicate in vitro in IECs established from iPSCs (TkD2) carrying inactive mutations in the *FUT2* gene. Our analysis of these cells showed that α(1,2) fucose was not required for the in vitro growth of the GI.2 and GII.1 genotypes of HuSaV. Some human noroviruses, such as GII.3, can be infected independently of α(1,2) fucose [10]. Continuous investigations using in vitro replication systems are required to elucidate whether α(1,2) fucose and other carbohydrate chains are involved in HuSaV infection.

During the preparation of this manuscript, Estes MK, Desdouits M, and their colleagues reported that HuSaV replicates in an in vitro system using human tissue-derived IECs [26]. They also noted that replication of GI.1 HuSaV in the above systems is independent of secretor status. Our present study is in general agreement with their findings and supports the utility of IECs established from human iPSCs in HuSaV research. In addition to GI.1 HuSaV, our study indicated that GI.2, GII.1, GII.3, and GIV.1 genotypes of HuSaV were also able to replicate in an in vitro system independently of IEC secretory status.

In summary, our study established an in vitro replication system for HuSaV using human IECs derived from iPSCs. Using this assay system, we confirmed that viral inactivation can be studied physiologically. We also showed that infection with at least the GI.2, GII.1, GII.3, and GIV.1 genotypes of HuSaV occurs in a fucose-independent manner.

## Figures and Tables

**Figure 1 viruses-15-01929-f001:**
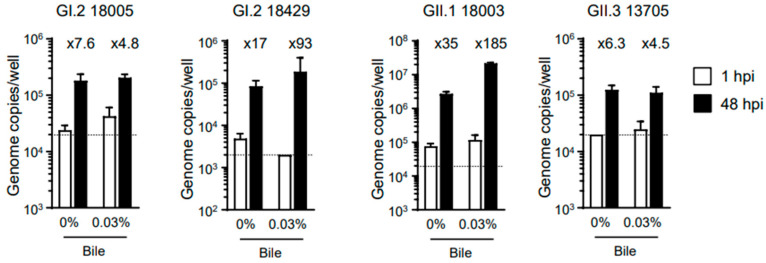
Requirement for bile during the in vitro replication of HuSaV. Monolayers of human iPSC-derived IECs were cultured with or without 0.03% porcine bile and then inoculated with 2 × 10^6^ genome equivalents of the indicated HuSaV genotypes. Inoculation and sampling were performed as described in the Methods section. Viral genome RNA was extracted from supernatants sampled at 1 or 48 hpi, and the genome equivalents were quantified by RT-qPCR. Each value is representative of at least three independent experiments and is shown as the mean ± SD from four wells of supernatants from each culture group. The mean fold changes in viral genome between the two time points are indicated above each black bar (48 hpi).

**Figure 2 viruses-15-01929-f002:**
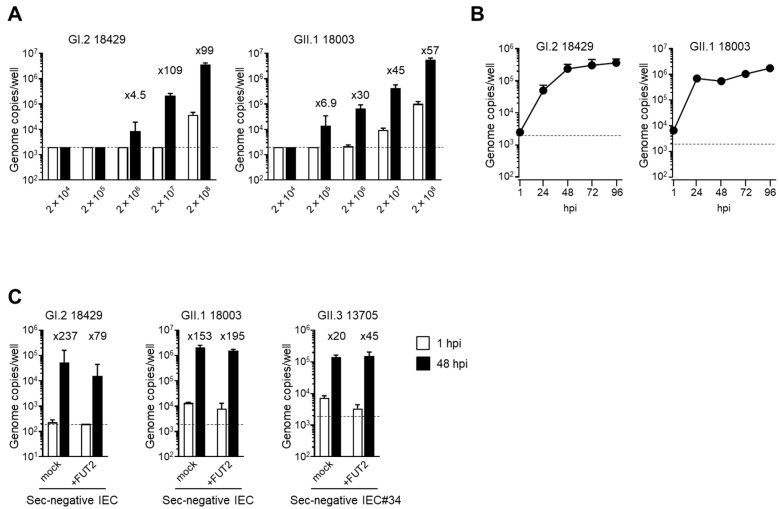
Replication of HuSaVs in IECs derived from human iPSCs. Monolayers of human iPSC-derived IECs were inoculated with the indicated genome equivalents (**A**) or 2 × 10^7^ genome equivalents (**B**,**C**) of the indicated HuSaV genotypes. Inoculation and sampling were performed as described in the Methods section. Viral genomic RNA was extracted from supernatants sampled at 1 or 48 h postinfection (hpi) (**A**,**C**) or the indicated times (**B**), and genome equivalents were then quantified by real-time PCR. Each value is representative of at least three independent experiments and is shown as the mean ± SD from four (**A**,**C**) or three (**B**) wells of supernatants from each culture group. The mean fold changes in the viral genome between the two time points are indicated above each black bar (48 hpi). Dashed lines represent the limit of detection.

**Figure 3 viruses-15-01929-f003:**
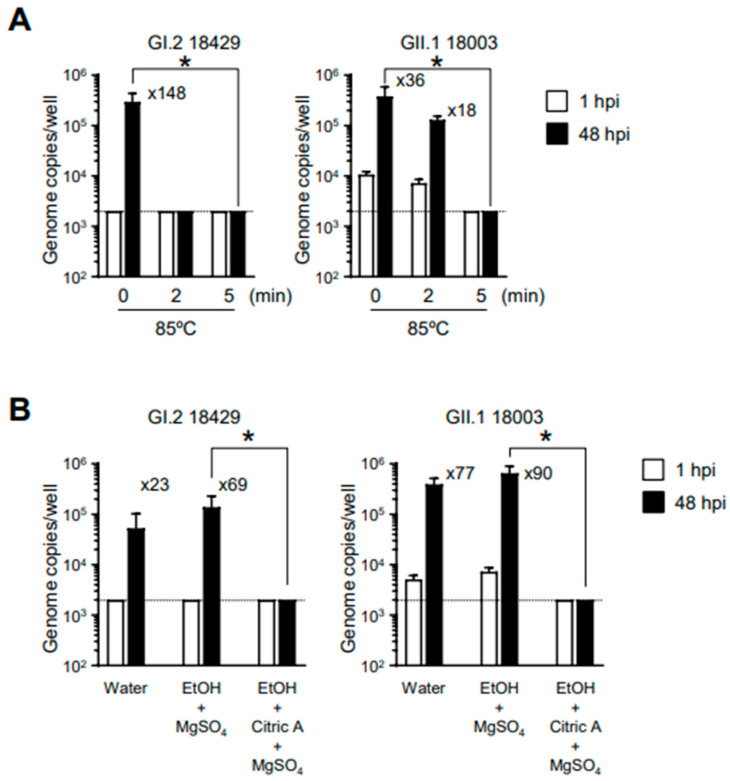
Inactivation of HuSaVs by heat and acid-ethanol treatments. GI.2 and GII.1 HuSaVs were incubated at 85 °C for the indicated times (**A**) or were suspended with 3 volumes of the indicated disinfectants at room temperature for 5 min (**B**). Monolayers of IECs were inoculated with 2 × 10^7^ (**A**), 5.3 × 10^6^ (**B**, GI.2), or 2.5 × 10^7^ (**B**, GII.1) genome equivalents of the indicated HuSaV genotypes. Viral genome RNA was extracted from both supernatants, and the genome equivalents were quantified by real-time PCR. Samples at 1 hpi were used as references. Each value is representative of three independent experiments and is shown as the mean ± SD from four wells of supernatants for each culture group. The mean fold changes in viral genome between the two time points are indicated above each black bar (48 hpi). EtOH, 70% ethanol; MgSO_4_, 0.05% magnesium sulfate; Citric A, 1% citric acid. * *p* < 0.05. Dashed lines represent the limit of detection.

**Figure 4 viruses-15-01929-f004:**
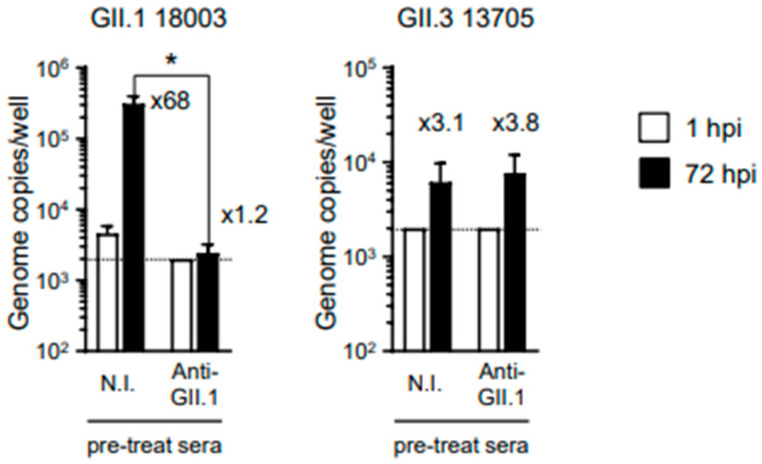
Neutralization of GII.1 HuSaV by antiserum induced with the immunization of GII.1 HuSaV VLPs. First, 2 × 10^7^ genome equivalents of GII.1 or GII.3 HuSaVs were incubated with 0.2 μL of serum prepared from non-immunized mice (N.I.) or anti GII.1 HuSaV antiserum at 37 °C for 90 min before inoculation into the prepared IECs. Viral genomic RNA was extracted from supernatants sampled at 1 or 72 hpi, and genome equivalents were then quantified by real-time PCR. Each value is representative of three independent experiments and is shown as the mean ± SD from six wells of supernatants from each culture group. The mean fold changes in viral genome between the two time points are indicated above each black bar (72 hpi). * *p* < 0.05. Dashed lines represent the limit of detection.

**Table 1 viruses-15-01929-t001:** List of human sapoviruses used in this study.

SaV Sample ID	Genotype	Titer (Genome eq./well)	Patient Age (mo)	Patient Gender	Area	Collection Date	NCBI Accession No.	Fold Viral RNA Increase in IEC#17
13396	GI.1	1.38 × 10^7^	16	Male	Gunma	May 2015	LC549540	–
13699	GI.1	1.34 × 10^8^	21	Male	Gunma	July 2015	LC549544	–
18002	GI.1	1.15 × 10^7^	11	Male	Kyoto	March 2019	–	–
18004	GI.1	3.52 × 10^7^	12	Male	Kyoto	March 2019	–	4.8
18363	GI.1	1.10 × 10^8^	11	Male	Kyoto	March 2019	–	–
18365	GI.1	9.54 × 10^6^	12	Male	Kyoto	March 2019	–	–
18529	GI.1	1.37 × 10^7^	17	Female	Shizuoka	January 2019	–	–
12968	GI.2	1.34 × 10^7^	24	Male	Gunma	January 2015	LC549536	–
13005	GI.2	1.14 × 10^6^	22	Female	Gunma	January 2015	LC549537	–
18005	GI.2	5.02 × 10^4^	18	Female	Kyoto	March 2019	–	11.8
18429	GI.2	2.11 × 10^7^	22	Male	Kyoto	March 2020	–	875.7
16811	GI.3	2.51 × 10^5^	61	Male	Gunma	June 2017	LC549642	–
13299	GII.1	9.86 × 10^6^	23	Male	Gunma	March 2015	LC549563	–
13385	GII.1	5.52 × 10^6^	12	Male	Gunma	May 2015	LC549539	–
18003	GII.1	1.01 × 10^8^	11	Male	Kyoto	March 2019	–	201.7
18547	GII.1	4.46 × 10^7^	14	Male	Shizuoka	July 2019	–	–
13705	GII.3	4.10 × 10^7^	25	Male	Gunma	July 2015	LC549545	20.6
13473	GIV.1	3.26 × 10^7^	59	Female	Gunma	July 2015	LC549542	3.0
14542	GIV.1	9.08 × 10^6^	19	Male	Gunma	February 2016	LC549552	13.9

–: less than 1.0.

## Data Availability

Not applicable.

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
