# Peer review of "Replication of Human Sapovirus in Human-Induced Pluripotent Stem Cell-Derived Intestinal Epithelial Cells"

_viruses, 2023, doi:10.3390/v15091929_

Round 1
Reviewer 1 Report
Matsumoto et al describe successful results obtained after testing whether several human sapovirus strains are able to replicate in iECs derived from iPSC. The paper is well written, and clearly described. The work is relevant and interest to the field, but several points should be addressed before publication:
MAJOR POINTS
- Supplementary figure 1 and table 1 should be shown in the manuscript. The authors conclude that bile acids were not always necessary, but data in sup fig 1 shows that all 4 tested stool samples (belonging to 3 different genotypes) do replicate in the absence of bile acids. This information is contradictory and should be addressed and further discussed in the paper.
- The authors should show whether viruses released from infected IECs cells were infectious or not. Were they able to further passage the viruses onto new cells?
- The authors should discuss whether their replication system was sensitive to the use of frozen stool samples versus fresh ones. Did freezing of stool samples afect infectivity?
- Standard material used for quantification in real time assays should be described. Was is the same for all genotypes?
- Figure 1. Indicate whether bile acids were included in the experiments showed. The meaning of dotted line should be described in the figure legend. Fold increase observed in each experiment should be added to the figures. It is unclear why Figure 1B does not contain error bars for GII.1 sample.
- Figure 1 It is unclear why GI.1 18429 infecting at 2x10^7 genome equivalents is detected below the cutoff threshold. Please discuss why.
- Figure 2A. It is surprising to see that GII.1 sample is resistant to heatint ag 85ºC for 2 minutes. Please further discuss this result.
- Figure 2. Meaning of * is unclear and should be described in more detail.
-The authors conclude that alcohol treatment could "completely" inactivate human sapoviruses (line 277). However, the system requires more than 10^6 genome copies/100 ul for replication. How can the authors be sure that viruses are completely inactivated? Please further discuss or clarify.
- The authors describe the generation of novel HuSaV GII.1 antiserum, but data of ELISA experiments performed to further characterize this antiserum should be reported. Otherwise it is unclear whether this antiserum is sensitive and specific. Is this antiserum specific fo GII.1 virions or can it also recognize other genotypes?
- Figure 3. Indicate viral doses used in these experiments. Were they similar for both genotypes? Are data comparable?
- Figure 3. The authors should describe what is "Normal" pre-treat sera... it is not clear. A control showing infectivity after using an unrelated antiserum should be included in the figure.
- Figure 3. The statistical analysis for GII.3 data is unexpected. Could the authors confirm this result?
MINOR POINTS
- Line 84. Complete reference should be added instead of doi.
- Line 97. Multiplicity of infection (MOI) should be described not only as genome equivalents per 100 ul of inoculum, but also as genome equivalents per surface area or numbr of infected cells.
Reviewer 2 Report
Using a real-time RT-PCR, the authors investigated infection and replication of six human sapovirus genotypes (GI.1, GI.2, GI.3, GII.1, GII.3, and GIV.1) in human iPSC-derived intestinal epithelial cells (IECs), the effects of porcine bile addition and glycosylation of α (1,2)-type fucose on IECs on several genotypes of human sapovirus infection and replication, inactivation of the virus by heating to 85 ℃ or 70% alcohol-0.05% magnesium sulfate with or without 1% citric acid, and inhibition of viral infection/replication on homologous and heterologous human sapovirus GII genotypes by mouse antiserum prepared with VLP antigens.
Two in vitro infection/propagation or replication systems have been reported for human sapovirus in these 2-3 years.
1) an infection/ propagation system using human duodenum derived HuTu80 cells (reference 10 and others)
2) an infection/replication system using organoids derived from multiple sites in the human small intestine (reference 19)
Thus, the present authors' report is likely the third, but including novel data (e.g., non requirement of bile for viral infection and replication, disinfection by acid alcohol, and neutralization by anti-serum).
In general, the title and text should use "replication" instead of "propagation”, because the authors confirm only the viral nucleic acid increased levels (up to several hundredfold), and
no data was shown for the detection of proliferating viruses at the protein level, and preparation of serial passaged virus stocks with high titer.
For all bar graphs showing comparisons with 1hpi, it would be easier to understand the results if the fold viral RNA increase values were inserted in the graphs (see reference 19).
Furthermore, it is not clear which experiments are bile-added or non-bile-added throughout.
For some of the experimental systems, there is a lack of important information to evaluate the experimental systems, such as the method of disinfectant removal/dilution after treatment, as well as the number of viral copies added in the neutralization assay.
In addition, the following revisions are necessary to aid the reader's understanding.
Lines 17−26
Abstract
Vague content. Revise to specifically include the content and novelty of this research.
Specifically, the following contents could be written.
1) Infectivity of multiple genotypes of human sapoviruses could be evaluated using intestinal epithelium derived from human iPS cells,
2) Biles and FUT2 gene were not essential for replication of these sapoviruses
3) Disappearance of infectivity with heating at 85°C for 5min, with ethanol containing citric acid and MgSO4, and by antiserum prepared against VLPs.
Introduction
Lines 35-37.
Information of Reference 3 should be updated, and add other reports of an increase in the detection frequency of human sapoviruses from acute gastroenteritis patients.
Lines 41-44
Since the preceding and following paragraphs do not connect well with this context and do not relate to the main purpose of this paper, I recommend deleting the statement "On the other hand,...".
Line 45 and others
State clearly what is “closely related”.
Line 49
Reference 10 states up to 6 log10-fold increase.
Lines 50-54
The authors emphasize the superiority of normal cells compared to cancer-derived cells.
I would request that the authors objectively describe the equivalence, advantages, and disadvantages of their system compared to those already reported.
1) Is it possible to generate viral particles, serial passage of propagating viruses, and preparation of high viral copy number virus stocks as reported in Reference 10? They reported viral RNA levels increased up to 6 log10 fold.
2) Does the present system of the authors have any advantage over the human intestinal enteroid system of Reference 19? They reported viral RNA levels increased up to 3 log10 fold.
Materials and Methods
Line 60
Based on the description in the lines 160-161, I would suggest revise this sentence as follows:
“The human iPSC-lines … established by The institute of Medical Science, The University of Tokyo (reference 8,11, and 12) were used in this study”.
Lines 81-88
Are the fecal specimens used in this viral replication study derived from acute gastroenteritis outpatients?
The article 10.1002/jmv.26934 should be added to the references.
Since 10.1002/jmv.26934 only describes a screening for sapovirus, description of Rota, noro, adeno, astro, parecho, and enteroviruses can be deleted.
GI.2 13005 is also described in 10.1002/jmv.26934. The origin of the specimens should be accurately described, including specimens that have been previously reported and analyzed to genotype, and other specimens that have been newly analyzed.
Lines 89-106
Is the 0.32mm2 used in Figure referring to the area of one well of a 96-well plate (0.32cm2)?
Alternatively, the authors can be changed the Y-axis of Figure to say per well or per µL of supernatant.
Lines 107-113
The reagents/kits used for real-time PCR, the reaction conditions, as well as the reference used for sapovirus genome quantitation should be added.
Also, add description for the dotted lines in the Figures.
Lines 114-129
Centrifugal conditions should be described in xg, not rpm.
Despite the detailed protocol described, there are no results at all.
It is mandatory to show at least an electron micrograph of the VLP in the supplement figure, because the authors describe that “VLP structure was confirmed”.
Lines 130-143
Coated with 1 µg/mL should be stated as volume used per well or how many ng per well.
The specific results of Antibody titer are not shown despite the detailed description of the measurement method. Show at least in the supplement figure.
Results
Line 159
Does Patients mean acute gastroenteritis patients?
Lines 160
“Harboring” can be omitted.
Line 163
Bile was not always necessary (Supplementary Figure 1).
The results in Figure 1-3 and Supplementary Table 1 should be clarified whether or not bile was added.
Line 178
Since the Figure 2B shows the experimental results at 5.3x106 and the supplement Figure 1 also shows the results at 2x106, I suggest removing the sentence for “, and stable replication was observed....”.
Line 180
The reviewer disagrees about reached a plateau at 48 hpi.
Both GI.2 and GII.1 viral RNA levels are gradually increased to 96 hpi.
Line 186
Does Reference 7 mention anything about human norovirus entry?
Lines 188 and190
What the authors confirm in this paper is the viral nucleic acid increased levels of up to several hundredfold.
“Infection and subsequent prolification”, and “prolification”, should be unified with “replication”.
Lines 194-195
Preparation of IEC#34 with lentiviral transduction of the FUT2 gene should be described in Materials and Methods.
Line 203
How about revise “transfer” to “an expression”?
Line 223
Inconsistent description between "Mixed with nine volumes of…” in line 223, and "… suspended with 3 volumes of" in the Figure 2 legends. The description should be standardized. Furthermore, the description of dilution/removal process after disinfectant treatment is missing. It is important to show that disinfectant has no toxicity on the cells.
How did the authors confirm that inactivating the infectivity of human sapovirus was not due to cytotoxicity caused by citric acid? To indicate precise control, the authors should indicate the data adding virus later to the disinfectant treated (diluted) solution.
The details should be described in the Materials and Methods section.
Line 233
Outdated citations on norovirus vaccine; add Reference 17 and beyond.
Lines 233-235
Recommend deletion.
The lack of momentum for vaccine development against sapoviruses may be due to that the frequency and severity of detection in gastroenteritis cases has been considered low.  Given the time frame for the norovirus vaccines history, the presence of a replication system in cultured cells is not necessarily relevant to their development.
Line 236
The authors' system is likely not a propagation system.
Lines 241-244
Details of the assay is unclear. It should be described in Materials and Methods.
What is the RNA copy number of the input sapovirus?
Since the measurement of antiserum titer of VLP-immunized mice is described in Materials and Methods, add the results of antiserum titer in Supplement Figure.
Line 245
“Cross-reactivity between different genotypes is not expected. ”
As with the other results, the comparison with norovirus should be discussed here, and references should be added.
Discussion
Line 258
Delete “Still”.
Lines 259-261
Although “an in vitro culture system resembling to human intestinal epithelium for human SaV may be important for the final evaluation of vaccines and therapeutics”, an efficient growth system for human sapoviruses already exists (Ref. 10). Therefore, this explanation is unreasonable. Revision is recommended.
Line 269
The authors' culture system uses monolayer cells. Please specify what constitutes a “more physiological environment”.
Line 270
It has not reached a plateau.
Lines 281-283
Vague statement. Delete or describe more specifically.
Line 284
Add reference(s).
Lines 285-286
Can in vitro growth (replication) system evaluate pathogenicity?
Lines 297-303
Why not mention GIV in here, which is considered in this paper and also has Fold viral RNA increase values in Supplementary Table 1?
Line 306
What does Physiologically mean?
Supplementary Table 1
Is titer (genome eq/ell) a typo?
viral RNA increase levels were from the condition with bile or not? Indicate in clearly.
Supplementary Figure 1
Reconfirm the units of the Y-axis.
It would be easier to see if the increased fold values with and without bile are inserted in the graph. Is there a statistical difference in viral nucleic acid copy number between GI.2 and GII.1 with and without bile?
Why the authors not considering GIV.1 14542 with 13.9 fold viral RNA increase despite including GI.2 18005 with 11.8 fold increase?
Supplementary Figure 2
Add a description and reference of the Norovirus GII.4 used.
Review and editing by native speakers is preferred.
Round 2
Reviewer 1 Report
The major points raised during the revision of the paper have been addressed.
Author Response
We all thank you for your peer review of our manuscript.
Reviewer 2 Report
The authors did not appear to have uploaded a legend for the supplemental figure. As a result, I was unable to verify the content and validity of the newly added figures. It is essential that this additional material be provided in order to complete the peer review.
The revised manuscript, references, and figures themselves seem to reflect the reviewers' points well, so please address this request.
Author Response
Please forgive us. I have uploaded a WORD file of the supplement figure legends in the reply.
